# A Revisit of Large-Scale Patterns in Middle Stratospheric Circulation Variations

**DOI:** 10.3390/e27040327

**Published:** 2025-03-21

**Authors:** Ningning Tao, Xiaosong Chen, Fei Xie, Yongwen Zhang, Yan Xia, Xuan Ma, Han Huang, Hongyu Wang

**Affiliations:** 1School of Systems Science, Beijing Normal University, Beijing 100875, China; ningningtao@mail.bnu.edu.cn (N.T.); xiayan@bnu.edu.cn (Y.X.);; 2Institute for Advanced Study in Physics and School of Physics, Zhejiang University, Hangzhou 310058, China; 3Data Science Research Center, Faculty of Science, Kunming University of Science and Technology, Kuming 650500, China; zhangyw@kust.edu.cn

**Keywords:** stratospheric variations, large-scale patterns, eigen microstate approach

## Abstract

Variations in stratospheric atmospheric circulation significantly impact tropospheric weather and climate. Understanding these variations not only aids in better prediction of tropospheric weather and climate but also provides guidance for the development and flight trajectories of stratospheric aircraft. Our understanding of the stratosphere has made remarkable progress over the past 100 years. However, we still lack a comprehensive perspective on large-scale patterns in stratospheric circulation, as the stratosphere is a typical complex system. To address this gap, we employed the eigen microstate approach (EMA) to revisit the characteristics of zonal wind from 70–10 hPa from 1980 to 2022, based on ERA5 reanalysis data. Our analysis focused on the three leading modes, corresponding to variations in the strength of the quasi-biennial oscillation (QBO) and the stratospheric atmospheric circulations in the Arctic and Antarctic, respectively. After filtering out high-frequency components from the temporal evolutions of these modes, a significant 11-year cycle was observed in the Antarctic stratospheric atmospheric circulation mode, potentially linked to the 11-year solar cycle. In contrast, the Arctic stratospheric atmospheric circulation mode showed a 5–6-year cycle without evidence of an 11-year periodicity. This difference is likely due to the timing of polar vortex breakdowns: the Antarctic polar vortex breaks up later, experiencing its greatest variability in late spring and early summer, making it more susceptible to solar radiation effects, unlike the Arctic polar vortex, which peaks in winter and early spring. The fourth mode exhibits characteristics of a Southern Hemisphere dipole and shows a significant correlation with the Antarctic stratospheric atmospheric circulation mode, leading it by about two months. We designed a linear prediction model that successfully demonstrated its predictive capability for the Antarctic polar vortex.

## 1. Introduction

The stratosphere, located approximately 10 to 50 km above the Earth’s surface and containing around 17% of the Earth’s atmospheric mass, is crucial to human life. As early as 2001, Baldwin pointed out that anomalies in stratospheric circulation can descend to influence the troposphere and serve as predictive signals for weather and climate [1]. Today, stratosphere processes—along with their coupling with the troposphere—play a significant role in sub-seasonal to seasonal weather predictions [2].

Our understanding of the stratosphere has evolved significantly over the past 100 years [3]. In the early 20th century, the existence of the stratosphere and the ozone layer was first discovered. With advancements in observational technology, an increasing number of stratospheric processes have been identified. Concepts and theories explaining these processes were subsequently developed, including the Brewer–Dobson circulation [4,5,6], the sudden stratospheric warmings (SSWs) [7,8,9,10], the quasi-biennial oscillation (QBO) [11,12,13,14,15], the wave–mean flow interaction [16,17,18], the parameterization of waves in climate models [19,20,21,22], the stratosphere–troposphere coupling [2,23], the concept of “mean age of air” [24], the chemical reactions of stratospheric ozone [25,26,27,28,29] and the cause of the Southern Hemisphere ozone hole [30,31,32,33,34,35,36,37]. These pioneering works have significantly extended our understanding of the complex dynamics of stratospheric variations.

One noteworthy observation during this period is that advances in theory have often been promoted by unexplained phenomena. However, none of these phenomena were anticipated by theory [3]. This is different from the development of physics, where theory can also predict phenomena. For example, the discoveries of black holes [38], gravitational waves [39], and the Higgs boson [40] were all guided by theoretical predictions.

This difference arises because the stratosphere, as a subsystem of the Earth’s atmosphere, is a complex system [41]. It is governed by intricate atmospheric dynamic processes, which are in turn regulated by atmospheric chemical processes. Waves and their interaction with the mean flow play a crucial role in driving these processes [17,42]. Moreover, these waves span a wide range of scales, from small-scale gravity waves to global-scale planetary waves [3]. The renowned physicist Philip W. Anderson’s statement “More is Different” vividly captures the essence of emergent behavior in complex systems [43]. He highlighted that as systems grow in scale and complexity, entirely new properties emerge—properties that cannot be understood by simply extrapolating from the behavior of individual components. Understanding such emergent behaviors requires fundamental research. Therefore, the stratosphere, as a complex system, demands the study of emergent large-scale climate patterns of its variations, which is as essential and fundamental as understanding the dynamics that govern it.

EMA is an efficient tool used to study collective behaviors, emergence, phase transitions, and evolution in complex systems. This approach has already been widely applied across various complex systems, including physical systems, biological systems, and climate systems [44] (See Appendix A). For example, it has been used to study equilibrium and non-equilibrium phase transitions [45,46,47,48,49]; to predict extreme El Niño events [50]; to investigate vegetation growth and evolution patterns in the Heihe River Basin [51]; to analyze the characteristics of stratospheric ozone distribution [52]; and to explore dominant patterns of brain activities [53]. The concept behind the EMA originates from the statistical physics of Gibbs, in which the states of all particles in a system at any given time are considered a microstate, and the microstates over a period of time are treated as an ensemble of the complex system [44]. By calculating the eigenvectors of the correlation matrix between these microstates, the system’s eigen microstates are obtained, which are uncorrelated with each other.

The most critical aspect of EMA is the definition of microstates within a system. This requires a detailed analysis based on the characteristics of the system and the problems being addressed, as well as the selection of appropriate variables (details of the EMA are provided in Section 2.2). Wind is a key variable for characterizing the stratosphere, as the QBO and the polar vortex are defined using wind-based indices. Furthermore, the stratosphere is part of near-Earth space, a transitional region where spacecraft and high-altitude vehicles operate. The development of near-Earth space vehicles is crucial for defense and military applications, while low-power near-Earth space vehicles can be employed for purposes such as meteorological monitoring, atmospheric environmental protection, and ground-based remote sensing. Understanding the variations of stratospheric wind is, therefore, vital for guiding the design and operation of these vehicles. Given that the zonal wind in the stratosphere is an order of magnitude stronger than the meridional wind, in this paper, we will use the EMA to study the emergent large-scale patterns of global stratospheric zonal winds. Although large-scale modes in the stratosphere have been studied for decades, it remains meaningful to investigate the modes of stratospheric zonal winds using EMA for two main reasons. First, by defining microstates more effectively through EMA, it becomes possible to uncover new modes that were previously unidentified. Second, the contributions and spatial patterns of existing modes may change over time.

The remainder of this paper is organized as follows: Section 2 introduces the data and methods. Section 3 presents the results, and Section 4 provides a summary.

## 2. Data and Methods

### 2.1. Data

This study utilizes global zonal wind and sea surface temperature data from the ERA5 (ECMWF Reanalysis v5) monthly reanalysis dataset [54], covering the period from 1980 to 2022, with a total of *M* = 43 × 12 = 516 temporal observations. The spatial resolution of ERA5 is 0.25° × 0.25° in latitude and longitude. For this study, we downsampled the data to a 1° × 1° resolution, where the value at each grid point represents the corresponding 1° × 1° cell, resulting in *N* = 360 × 181 = 65,160 discrete grid points. ERA5 provides wind speed data at 37 different pressure levels, ranging from the surface (1000 hPa) to the top of the stratosphere (1 hPa). For this study, we selected four levels—70, 50, 30, and 10 hPa—covering the lower to middle stratosphere. The sudden stratospheric warmings (SSWs) refer to the breakdown of the polar vortex, accompanied by a rapid descent and warming of the air in polar latitudes [10]. The zonal–mean zonal wind at 60° latitude is commonly used as an indicator of the strength of the polar vortex and to determine whether an SSW event has occurred. Therefore, we defined the Arctic polar vortex index and Antarctic polar vortex index at each pressure level as the zonal mean zonal wind at 60° N and 60° S for their respective pressure levels [55]. To further analyze the data, we removed the seasonal cycle by subtracting the monthly mean values across all years, resulting in the deseasonalized Arctic polar vortex index (NPVI) and deseasonalized Antarctic polar vortex index (SPVI). We also used the QBO index defined as the average zonal wind between 5° N and 5° S [56,57]. Additionally, we used the Niño 3.4 index provided by NOAA (National Oceanic and Atmospheric Administration) [58]. The Niño 3.4 index represents the average sea surface temperatures (SST) anomalies across the Niño 3.4 region (5° N–5° S, 170° W–120° W).

### 2.2. Eigen Microstate Approach

The EMA has been recently developed for studying collective behaviors, emergence, and phase transition of complex systems [44,45]. Here, we will use it to study the stratospheric zonal wind. We take a certain pressure level in the stratosphere as a complex system, which is divided into N=360×181=65160 grid points with a spatial resolution of 1° × 1°. With a monthly resolution spanning from 1980 to 2022, we obtained M=516 temporal observations.

Due to the influence of solar radiation, the stratospheric zonal wind exhibits pronounced seasonality. However, our primary interest lies in the behavior of the zonal wind beyond its seasonal variations. Therefore, before applying the EMA, we must first remove the seasonal component from the zonal wind data:

Let $u_{i}^{y}\left(m\right)$ denote the zonal wind speed at grid point *i* in the *m*th month of year *y*, where i∈[0,N), m∈[0,12), and $y\in [0,N_{y})$, with $N_{y}$ representing the number of years. Let ${\bar{u}}_{i}^{m}=\frac{1}{N_{y}}\sum_{y=0}^{N_{y}-1}u_{i}^{y}\left(m\right)$ represent the average zonal wind at grid point *i* in the *m*th month.

To deseasonalize the zonal wind, we calculate the following:$${\tilde{u}}_{i}^{y}\left(m\right)=u_{i}^{y}\left(m\right)-{\bar{u}}_{i}^{m}$$
where ${\tilde{u}}_{i}^{y}\left(m\right)$ represents the deseasonalized zonal wind at grid point *i*, indicating fluctuations relative to the seasonal trend.

Let t=y×12+m, then the deseasonalized zonal wind ${\tilde{u}}_{i}^{y}\left(m\right)$ at grid point *i* can be written as ${\tilde{u}}_{i}\left(t\right)$, where t∈[0,M). Let $std\left({\tilde{u}}_{i}\right)=\sqrt{\frac{1}{M}\sum_{t=0}^{M-1}{\left[{\tilde{u}}_{i}\left(t\right)\right]}^{2}}$ represent the standard deviation of the deseasonalized zonal wind at grid point *i*.

Due to the spatial heterogeneity of zonal winds at different latitudes and longitudes in the Earth system, the fluctuation amplitude varies across grid points. Therefore, it is necessary to make the fluctuations at each grid point dimensionless by dividing them by their standard deviations. Consequently, we can define the system’s microstates as follows:$$S\left(t\right)=\left[\begin{array}{c} \frac{{\tilde{u}}_{0}\left(t\right)}{std({\tilde{u}}_{0})}\\ \cdots \\ \frac{{\tilde{u}}_{i}\left(t\right)}{std({\tilde{u}}_{i})}\\ \cdots \\ \frac{{\tilde{u}}_{N-1}\left(t\right)}{std({\tilde{u}}_{N-1})} \end{array}\right]$$

Taking all microstates at various times, we obtain an N×M ensemble matrix A with the following elements: $A_{i}\left(t\right)=\frac{1}{\sqrt{C_{0}}}S_{i}\left(t\right)$, where $\sqrt{C_{0}}=\sum_{i=0}^{N-1}\sum_{t=0}^{M-1}S_{i}^{2}\left(t\right)$.

The spatial correlation between grid points can be calculated as $K_{ij}=\sum_{t=0}^{M-1}A_{i}\left(t\right)A_{j}\left(t\right)$, resulting in the spatial correlation matrix $K=AA^{T}$. The eigenvectors of the matrix K form a unitary matrix $U=\left[\begin{array}{c} U_{1},U_{2},...,U_{N} \end{array}\right]$. Similarly, the temporal correlation between microstates can be calculated as $C_{tt^{\prime }}=\sum_{i=0}^{N-1}A_{i}\left(t\right)A_{i}\left(t^{\prime }\right)$, resulting in the temporal correlation matrix $C=A^{T}A$. The eigenvectors of the matrix C form a unitary matrix $V=\left[\begin{array}{c} V_{1},V_{2},...,V_{M} \end{array}\right]$.

Mathematically, the eigendecomposition of the correlation matrices is equivalent to performing singular value decomposition (SVD) on the ensemble matrix $A=U\cdot \Sigma \cdot V^{T}$, where $\Sigma =\mathrm{diag}(\sigma_{1},\sigma_{2},...,\sigma_{R})$, and R=min(N,M) is the rank of the ensemble matrix, with $\sigma_{I}\left(I=1,...,R\right)$ being the singular values arranged in descending order. U and V are unitary matrices composed of the eigenvectors corresponding to the spatial correlation matrix K and the temporal correlation matrix C, respectively. Based on the SVD, the ensemble matrix A can be expressed as follows:$$A=\sum_{I=1}^{R}\sigma_{I}A_{I}^{e}$$
where $A_{I}^{e}=U_{I}⊗V_{I}$, with ${\left(A_{I}^{e}\right)}_{it}=U_{iI}V_{tI}$. The EMA decomposes the spatiotemporal evolution of a complex system into summations of eigenmodes with different weights, where $U_{I}$ and $V_{I}$ represent the spatial pattern and temporal evolution of the *I*th eigen microstate (EM), respectively. Since the definition of the ensemble matrix ensures that $\sum_{I=1}^{R}\sigma_{I}^{2}=1$, the contribution of each eigen microstate can be represented by $W_{I}=\sigma_{I}^{2}$. A larger contribution indicates that the EM contributes more to the original ensemble matrix.

Sometimes, the fluctuation amplitudes of the system may be affected by seasonality, which is reflected in the temporal evolutions as different amplitudes of $V_{I}\left(t\right)$ in different months. To describe the relationship between the fluctuation amplitudes and the month, we can define the seasonal variance of the mode in the *m*th month as follows:$$Var_{m}=\sum_{t\in S_{m}}V_{I}{\left(t\right)}^{2}$$
where $S_{m}$ is the set of dates that belong to month *m*. Since $V_{I}$ is a unit vector, we can deduce that $\sum_{m=1}^{12}Var_{m}=1$.

Reversing the signs of both the spatial pattern $U_{I}$ and temporal evolution $V_{I}$ (i.e., multiplying by −1) simultaneously does not affect the results. For ease of comparison, all eigen microstate results in this paper have been adjusted to ensure that the directions of $U_{I}$ and $V_{I}$ obtained at different pressure levels are aligned.

### 2.3. Temporal Correlation Coefficient

The temporal correlation coefficient measures the degree of linear correlation between two time series, *X* and *Y*. It is defined as follows:$$C_{XY}=\frac{\frac{1}{M}\sum_{t=0}^{M-1}\left(X_{t}-\bar{X}\right)\left(Y_{t}-\bar{Y}\right)}{\sqrt{\frac{1}{M}\sum_{t=0}^{M-1}{\left(X_{t}-\bar{X}\right)}^{2}\frac{1}{M}\sum_{t=0}^{M-1}{\left(Y_{t}-\bar{Y}\right)}^{2}}}=\frac{\sum_{t=0}^{M-1}\left(X_{t}-\bar{X}\right)\left(Y_{t}-\bar{Y}\right)}{\sqrt{\sum_{t=0}^{M-1}{\left(X_{t}-\bar{X}\right)}^{2}\sum_{t=0}^{M-1}{\left(Y_{t}-\bar{Y}\right)}^{2}}}$$
$X_{t}$ and $Y_{t}$ represent the values of time series *X* and *Y* at time *t*, respectively; $\bar{X}$ and $\bar{Y}$ denote the mean values of time series *X* and *Y*, respectively.

The value of $C_{XY}$ ranges from −1 to 1, where a coefficient of 1 signifies perfectly identical linear trends between the series, while -1 indicates completely opposite trends.

In certain scenarios, time-lagged correlation coefficients are utilized to evaluate predictive performance between time series *X* and *Y*. Assuming *X* leads *Y* (τ>0), the time-lagged correlation coefficients are defined as follows:$$C_{XY}\left(\tau \right)=\frac{\frac{1}{M-\tau }\sum_{t=\tau }^{M-1}\left(X_{t-\tau }-\bar{X}\right)\left(Y_{t}-\bar{Y}\right)}{\sqrt{\frac{1}{M-\tau }\sum_{t=\tau }^{M-1}{\left(X_{t-\tau }-\bar{X}\right)}^{2}\frac{1}{M-\tau }\sum_{t=\tau }^{M-1}{\left(Y_{t}-\bar{Y}\right)}^{2}}}$$$$\begin{array}{cc} & =\frac{\sum_{t=\tau }^{M-1}\left(X_{t-\tau }-\bar{X}\right)\left(Y_{t}-\bar{Y}\right)}{\sqrt{\sum_{t=\tau }^{M-1}{\left(X_{t-\tau }-\bar{X}\right)}^{2}\sum_{t=\tau }^{M-1}{\left(Y_{t}-\bar{Y}\right)}^{2}}} \end{array}$$

To assess the statistical significance of the correlation coefficient, a significance test is often required. Traditional significance tests for correlation coefficients assume that the data are independent and follow a Gaussian distribution [59]. However, this assumption is clearly unsuitable for climate time series due to their autocorrelations and inherent distributions. Therefore, it is necessary to design a null model tailored to the characteristics of climate time series for significance testing [60].

For instance, when calculating the correlation coefficient between EM2 and the global sea surface temperature (SST) series (in Section 3.2), we generated synthetic surrogate data from the EM2 and SST series individually using the Amplitude adjusted Fourier transform (AAFT) method [61,62] and computed the correlation coefficients 1000 times. The AAFT surrogate not only preserves the distribution of the series but also maintains its power spectra. The 90th and 10th percentiles of these 1000 samples were then used as the positive and negative thresholds for the correlation coefficients, respectively. Correlation coefficients exceeding these thresholds are considered statistically significant.

### 2.4. Linear Prediction Model

To quantitatively evaluate the predictive capability of EM4 for the Antarctic polar vortex in Section 3.4, we designed a linear prediction model. This model uses the temporal evolution sequence of EM4 to predict the (SPVI) (see Section 2.1). The prediction model is expressed as follows:SPVI(t)=a×EM4(t−2)+b+R(t)

Here, *a* and *b* are parameters learned through training, and R(t) represents the residual.

We trained the model using only data prior to the prediction time. For example, when predicting SPVI at time *t*, data from time 0 to t−1 were used as the training set. The training process involved deriving the eigen microstates from the training set, identifying the fourth mode and its temporal evolution, and utilizing the SPVI prior to time *t* with EM4’s temporal evolution to learn parameters *a* and *b*. Then the trained model, along with the value of EM4’s temporal evolution at t−2, was used to predict SPVI at time *t*. The reason for using a two-month lag in this prediction model will be detailed below in Section 3.4).

### 2.5. Spectral Peak Significance Test

The spectral significance test is used to assess whether a detected spectral peak in the power spectrum of the signal is statistically significant or could solely originate from (correlated) random noise.

The significance of a spectral peak is tested by comparing it against a red noise spectrum of a first-order autoregressive process fitted to the data. To assess the statistical significance of a spectral peak, we focus on the ratio of variances between the power spectrum of the signal, denoted as $\phi_{1}$, and the corresponding red noise power spectrum $\phi_{0}$, which is fitted from data. This ratio can be tested using the F-statistic:$$F=\frac{S_{1}^{2}}{S_{0}^{2}}$$
where $S_{1}^{2}$ and $S_{0}^{2}$ are the variances of the signal spectrum and the red noise spectrum, respectively.

For the signal $y_{1}\left(t\right)$, its autocorrelation function, $\gamma_{1}\left(\tau \right)$, and its power spectrum, $\phi_{1}\left(\omega \right)$, are Fourier transforms of each other. For red noise, we use the discrete red noise spectrum developed by Gilman et al. (1963) [63] where its unnormalized form is given by the following:$$\phi_{0}\left(\omega \right)=\frac{1-\rho^{2}}{1-2\rho \cos\left(\frac{h\pi }{N/2}\right)+\rho^{2}}$$
where h=0,1,2,⋯,N/2, N is the length of the time series and ρ is the lag-1 autocorrelation of the time series and we use the lag-1 autocorrelation from the signal $y_{1}\left(t\right)$ as its estimate.

The F-statistic requires the degrees of freedom $\nu_{1}$ for the numerator and $\nu_{0}$ for the denominator. For red noise $\nu_{0}$, it is typically assumed to be a large number. For the real-time signal $\nu_{1}$, we use $N/M^{*}$ as its estimate, where *N* is the length of the time series and $M^{*}$ is the number of independent spectral estimates. For the discrete power spectrum, $M^{*}=N/2$, and so $\nu_{1}=2$.

The steps for performing the spectral peak significance test are as follows:Calculate the power spectrum of the signal.Estimate the power spectrum of red noise based on the signal’s lag-1 autocorrelation.Calculate the ratio of the signal’s power spectrum to the red noise power spectrum.Perform a significance test on the ratio using the F-statistic.

## 3. Results

The stratosphere extends approximately from the tropopause at 70 hPa to the stratopause at 0.1 hPa. However, 99% of the atmospheric mass is concentrated below 10 hPa [64]. In addition, this study focuses on the stratosphere, aiming to minimize the influence of adjacent layers on mass continuity. To reduce the impact of the boundary layers near the troposphere and mesosphere, we did not select the layers close to these boundaries, such as 100 hPa or 1 hPa. Therefore, our study focuses on the following four representative layers: 70, 50, 30, and 10 hPa.

Figure 1 shows the contributions of the top four eigen microstates at 70, 50, 30, and 10 hPa. Together, they contribute about 40–50% of the variability in the stratospheric zonal winds. The first eigen microstate (EM1) has the highest contribution, around 20%. The second to fourth eigen microstates (EM2 to EM4) contribute approximately 13%, 9%, and 7% of the variance, respectively.

### 3.1. EM1 and Its Relationship with QBO

Figure 2a–d show EM1’s spatial patterns at four different pressure levels. Figure 2e–h depict the corresponding temporal evolutions (black solid lines), while Figure 2i–l present their power spectral densities. EM1’s spatial patterns across all pressure levels exhibit similar characteristics, with a strong correlation in zonal winds within the tropical stratosphere, approximately between 20° N and 20° S. The power spectra of EM1 reveal a significant cycle of around 2.4 years, indicating that the zonal winds in the tropical stratosphere reverse direction approximately every 28 months.

The QBO is a quasi-periodic oscillation phenomenon in the tropical stratosphere, where easterlies and westerlies alternately reverse, with an average cycle of about 28 months. These alternating easterlies and westerlies descend downward at a rate of about 1 km per month [15]. Thus, it can be inferred that EM1 is a mode associated with the QBO. The QBO was discovered in the early 1960s through observations of stratospheric zonal winds at Canton Island, which revealed a 2-year round oscillation [11,12]. Subsequently, theories were developed to understand this phenomenon [13,14], explaining that the QBO is a wave-driven circulation. The driving waves include small-scale gravity waves, global-scale Kelvin waves, Rossby waves, as well as mixed Rossby and gravity waves [65,66,67,68,69,70]. Although the mechanisms of QBO are known, the relative importance of these driving waves remains uncertain. State-of-the-art numerical models are still unable to represent QBO precisely, particularly during QBO disruption events [71,72].

The first QBO disruption occurred in 2015–2016. An anomalous upward displacement of westerly winds from 30 hPa to 15 hPa interrupted the normal downward propagation of the easterly phase. This disruption was unprecedented since observations began in 1953 [71]. Coincidentally, a similar QBO disruption occurred again four years later, during the winter of 2019–2020 [72]. Many studies have analyzed the causes of these two QBO disruptions, but the mechanisms behind their occurrence remain a topic of debate [73,74,75]. Some research suggests that, against the backdrop of global warming, QBO disruptions may become increasingly frequent [76].

The importance of the QBO is self-evident due to its wide-ranging impacts. For instance, the QBO can influence the strength of the polar vortex, as described by the Holton–Tan relationship [77,78,79,80]. Randel et al. used singular-value decomposition and regression analyses to identify QBO signals in ozone and nitrogen dioxide data in the stratosphere [81]. Rao and Yu et al. explored the climatology and trends of the northern winter stratospheric residual mean meridional circulation, which are influenced by the QBO, ENSO, and solar cycles [82]. Moreover, the QBO can also impact jet streams [83,84] and the Madden–Julian oscillation (MJO) [85]. A recent review article summarized four pathways of QBO teleconnections, providing a comprehensive overview of its broader impacts [86].

Currently, the QBO index is primarily used to represent the state and phase of the QBO and to measure its impact by correlating the index with climate variables in other regions or through composite analyses [56,57,87,88].

Selecting an appropriate QBO index is crucial for obtaining reliable research results. The most widely used QBO index is based on radiosonde observations collected in Singapore (1° N, 104° E) and compiled by the Free University of Berlin [89]. However, observations from a single meteorological station are insufficient to fully capture the QBO phenomenon due to its regional heterogeneity. The strength of the QBO signal diminishes as latitude increases [90]. A QBO index constructed from a larger set of data can better reflect the overall characteristics of the QBO. With advancements in observations and climate modeling, stratospheric reanalysis data have become increasingly accurate, enabling the construction of more precise QBO indices. Some studies have defined the QBO index using the average zonal wind between 5° N and 5° S [56,57]. Considering the descending nature of the QBO, the phase at different pressure levels is also an important feature. Other studies have employed EOF (empirical orthogonal function)-based methods to measure both the amplitude and phase properties of the QBO [91].

We compared EM1’s temporal evolutions with the commonly used QBO index based on zonal wind averages from 5° N to 5° S (the orange dashed line in Figure 2e–h). The two align well, with a correlation coefficient exceeding 0.88. However, slight differences are observed during the QBO’s maximum easterly and westerly phases, as well as during periods of QBO disruption (green-shaded areas). Notably, EM1’s spatial patterns (Figure 2a–d) indicate that the zonal extent of the QBO is broader than 5° N to 5° S. This is because the EMA captures the common features (emergent characteristics) across all spatial grid points. By normalizing each grid point by its own standard deviation, the heterogeneity between spatial grid points is reduced, enabling the detection of even very weak QBO signals at higher latitudes. This suggests that EM1’s temporal evolutions, derived from global zonal wind using the EMA, provide more detailed information about the QBO’s distribution. Additionally, EM1’s spatial patterns clearly illustrate the range and intensity distribution of the QBO phenomenon.

### 3.2. EM2 and the Arctic Polar Vortex Variations

The stratospheric polar vortex is a large-scale cyclonic circulation located in the stratosphere over the polar regions. It forms in autumn as temperatures in the polar stratosphere drop and weaken or break down in spring [92]. The polar vortices play a critical role in stratospheric and tropospheric weather patterns, including their influence on the ozone layer [93,94] and sudden stratospheric warmings (SSWs) [1,10,92,95,96,97,98,99].

SSWs are some of the causes of extreme cold events, which have a significant impact on human life. Extensive research has been conducted on the causes, impacts, and predictability of SSWs. For instance, Yu and Cai revealed the linkage between SSWs and surface cold-air outbreaks (CAOs) [100]. Rao et al. demonstrated the deterministic predictable limit of the February 2018 major SSW using a climate system model [101]. Bao et al. classified the tropospheric precursor patterns of SSW [102], while Ma et al. analyzed two possible causes of the rare Antarctic SSW in 2019 [103].

Figure 3a–d show EM2’s spatial patterns at four different pressure levels. They exhibit similar characteristics at each pressure level, primarily concentrated in the mid-to-high latitudes of the Northern Hemisphere. Specifically, strong westerlies are present in the 60° N–90° N region, while easterlies dominate the 30° N–50° N region. In Figure 3e–h, the gray solid lines represent EM2’s temporal evolutions, and the orange dashed lines represent the deseasonalized Arctic polar vortex index (NPVI). The significant overlap of these lines, with correlation coefficients exceeding 0.92, indicates that EM2 represents a mode describing variations of the Arctic polar vortex.

EM2’s temporal evolutions exhibit many “spikes”, suggesting that the intensity of the Arctic polar vortex fluctuates sharply. To explore the relationship between the severity of these fluctuations and the months of the year, we introduced the seasonal variance metric (see Section 2.2) from the EMA, with the results shown in Figure 3i–l. The figure shows that the strongest fluctuations in the Arctic polar vortex occur during the winter and early spring (January–March), which is when the strongest fluctuations in the Arctic polar vortex occur. In contrast, fluctuations are minimal in summer, as the Arctic polar vortex has already collapsed by that time [104].

Figure 4a–d show the power spectral density of EM2, revealing that EM2 contains not only strong high-frequency signals on an annual scale but also low-frequency trends. By applying a 12-month moving average to EM2, the high-frequency signals are filtered out, resulting in the black solid lines shown in Figure 3e–h. Figure 4e–h display the power spectral density of these smoothed signals. After filtering out the high-frequency signals, EM2 exhibits clear low-frequency trends, with significant cycles around 1.5 years and 2.4–2.6 years in its power spectrum.

We know that the 2.4–2.6-year cycle originates from the influence of the QBO. The QBO affects the Arctic polar vortex through the Holton–Tan Effect, where the polar vortex in the Northern Hemisphere’s winter is strengthened (weakened) when the QBO at 50 hPa is in its westerly (easterly) phase [77,105]. But the origin of the approximately 1.5-year cycle remains unclear. Given that stratospheric circulation is driven by wave-driven residual circulation, and that sea surface temperatures significantly impact these waves, we hypothesize that the 1.5-year cycle of the Arctic polar vortex originates from oceanic influences. To investigate this, we first applied a 12-month smoothing to EM2’s temporal evolutions to remove high-frequency signals and then calculated the correlation coefficients with global SST. Since the influence of SST on the Arctic polar vortex may be delayed, we calculated the correlation coefficients by leading the SST time series by 0–4 months ahead of EM2, selecting the maximum absolute correlation coefficient to generate the correlation map between SST and EM2. The results are shown in Figure 5a–d. It can be seen that SST in the central and western Pacific are significantly negatively correlated with EM2, especially at 70 hPa and 50 hPa. The distribution of correlation coefficients exhibits the characteristic horseshoe pattern associated with ENSO-related sea surface temperatures.

The sea surface temperatures in the central and western Pacific are related to ENSO, the most well-known climate phenomenon globally, which has widespread impacts on the world’s climate [106]. Research has shown that during El Niño events, the Brewer–Dobson circulation strengthens, which enhances the upward propagation of planetary waves and subsequently weakens the polar vortex in both hemispheres, while the opposite occurs during La Niña events [107]. Figure 6a–c show the lead–lag correlation between EM2 and the Niño 3.4 index, indicating a significant negative correlation, with ENSO leading EM2 by about four months.

By using Monte Carlo singular spectrum analysis (MC-SSA), Jevrejeva et al. [108] revealed that the periodic signals within ENSO can influence Northern Hemisphere winter climate variability, with a phase lag of approximately three months. Jevrejeva et al. speculated that these signals are most likely transmitted via the stratosphere, with the Arctic oscillation (AO) mediating their propagation through the coupled stratospheric and tropospheric circulation variability, which facilitates vertical planetary wave propagation. This may account for the observed four-month lag in the correlation between ENSO and EM2. Additionally, the power spectrum of the Niño 3.4 index (Figure 6d) also shows a significant 1.5-year cycle. Therefore, the 1.5-year cycle observed in the power spectrum of EM2 may be attributed to the influence of ENSO.

In addition to the 1.5-year cycle and the QBO cycle, the power spectrum of EM2 also shows a significant cycle of approximately 5–6 years at all pressure levels after applying a 36-month smoothing, as shown in Figure 4i–l. Similarly, we applied a 36-month smoothing to both the temporal evolutions of EM2 and SST, then calculated the correlation coefficients, as shown in Figure 5e–h. After this smoothing, there was a strong positive correlation between the North Pacific SST and EM2. Numerous studies have examined the interaction between North Pacific SST and the Arctic polar vortex. Some research studies indicate that an increase in North Pacific SST weakens the Aleutian Low, thereby reducing the upward flux of planetary waves and strengthening the polar vortex [109,110]. Other studies suggest that variations in Arctic stratospheric circulation can also influence North Pacific SST, triggering the Victoria mode, which in turn influences ENSO [111]. However, the 5–6-year cycle characteristics of the Arctic polar vortex have been scarcely mentioned in existing research, making this topic worthy of more in-depth and detailed study.

The observed 5–6-year cycle could potentially be associated with the 11-year solar sunspot cycle. Many natural terrestrial phenomena are influenced by the 11-year solar sunspot cycle, such as volcanic activity. In recent years, a number of studies have been published on the relationship between solar activity and volcanic activity [112,113]. The general conclusion from these studies is that volcanic activity is modulated by sunspots and associated space weather processes, with the strongest relationship observed during two phases of the 11-year solar cycle, near the minimum and maximum of sunspot activity. The proposed physical mechanism involves the forcing of electrical current systems between the ionosphere and the upper lithosphere, driven by solar X-rays and UV radiation during solar maxima (including extreme events such as solar flares) and by galactic cosmic rays (GCRs) during solar minima [113,114,115]. However, this kind of proposed physical linkage still needs to be further studied.

The impact of sunspot maxima and minima on terrestrial activity may potentially give rise to the observed 5–6-year cycle. However, due to the limited availability of volcanic eruption data, as well as the variation in eruption locations and magnitudes, it is difficult to achieve statistical significance within a 5–6-year time frame. Consequently, studies on volcanic eruption cycles often focus on longer periods [116,117].

Further research is needed to investigate and verify the relationship between the 5–6-year cycle and solar sunspot activity.

### 3.3. EM3 and the Antarctic Polar Vortex Variations

Figure 7a–d show EM3’s spatial patterns at four different pressure levels, with similar characteristics across all levels, primarily featuring strong westerlies near 60° S. In Figure 7e–h, the gray solid lines represent EM3’s temporal evolutions, while the orange dashed lines represent the deseasonalized Antarctic polar vortex index (SPVI). These lines overlap significantly, with correlation coefficients exceeding 0.88, indicating that EM3 is a mode reflecting variations in the Antarctic polar vortex. The black solid line represents the 12-month smoothed temporal evolutions, revealing that, in addition to high-frequency variations, EM3 also exhibits low-frequency variation characteristics. Figure 8e–h show the power spectral density of the 12-month smoothed temporal evolutions, revealing two significant cycles of approximately 1.8–2 years and around 2.6 years. The 1.8–2-year cycle is likely influenced by ENSO, while the 2.6-year cycle continues to be affected by the QBO. Although the impact of the QBO on the Antarctic polar vortex is not as well-known as the Holton–Tan Effect, numerous studies have observed the influence of the QBO on the Southern Hemisphere. For example, the QBO can influence tropical convection, generating Rossby wave trains that propagate southward to high latitudes in the Southern Hemisphere, affecting Antarctic sea ice [118]; during the QBO easterly phase, the circumpolar westerlies around Antarctica tend to slow down [119].

After applying a 36-month smoothing to temporal evolutions of EM3, its power spectral density was calculated, as shown in Figure 8i–l. Unlike EM2, EM3 does not exhibit a 5–6-year cycle; instead, it shows a cycle of approximately 11 years. Why is the 11-year cycle not significant in the Arctic polar vortex, while it is prominent in the Antarctic polar vortex? The 11-year cycle may possibly be linked to the 11-year sunspot cycle. Figure 7i–l display the seasonal variance of EM3, revealing that the strongest fluctuations in the Antarctic polar vortex occur in the late spring to early summer of the Southern Hemisphere (October to December), which also corresponds to the months with the greatest variability in the Antarctic polar vortex [120,121]. During the polar night of each hemisphere’s winter, the polar vortex is shielded from direct solar radiation, making it challenging for the sunspot cycle to exert a direct influence. Moreover, the breakup of the polar vortex in Antarctica occurs later than that of the Arctic [122]. However, the physical mechanisms behind this phenomenon require further analysis.

### 3.4. EM4: The Southern Hemisphere Dipole Mode

Figure 9a–d show EM4’s spatial patterns at four different pressure levels, while Figure 9e–h present its Antarctic projection. A typical feature of EM4’s spatial patterns is the dipole pattern in zonal winds, divided by the 60° S latitude, with contrasting winds to the north and south of this parallel. As altitude increases, a wave-1 pattern emerges across the eastern and western hemispheres, closely resembling the influence of planetary wave-1 on the Southern Hemisphere. In Figure 9i–l, the gray solid lines represent EM4’s temporal evolutions, while the black solid lines indicate the 12-month smoothed temporal evolutions, which also exhibit low-frequency variations. In the mid-to-lower stratosphere (70 and 50 hPa), the 12-month smoothed temporal evolutions show a significant correlation with the Niño 3.4 index with the latter leading by approximately 3 months. The correlation coefficients are 0.51 and 0.50 respectively. The orange dashed lines in Figure 9i,j represent the 12-month smoothed Niño 3.4 index.

In the mid-to-lower stratosphere (70 and 50 hPa), the power spectral density of EM4 reveals significant cycles of approximately 1.8 years, 2.5–2.6 years, and 3.8 years after a 12-month smoothing (Figure 10e,f), along with a 5.4–5.5 year cycle after a 36-month smoothing (Figure 10i,j). In contrast, these cycles vary in the middle stratosphere (30 and 10 hPa) (Figure 10g,h,k,l). The seasonal variance of EM4, shown in Figure 9m–p, indicates that the amplitude of EM4 is strongest during the winter and spring in the Southern Hemisphere.

We observed an interesting phenomenon: a leading connection between EM4 and EM3. Figure 11a–d display the lead–lag correlation between EM4 and EM3, where Time Lag < 0 indicates that EM4 leads EM3. The analysis reveals a significant correlation, with a correlation coefficient of around 0.4, when EM4 leads EM3 by two months. As we know, EM3 is a mode that reflects variations in the strength of the Antarctic polar vortex. This suggests that EM4 may have predictive significance for variations in the Antarctic polar vortex.

To quantitatively evaluate the predictive capability of EM4 for the Antarctic polar vortex, we designed a linear prediction model (see Section 2.4). The prediction results are shown in Figure 12. The model performs best at 70 hPa and 50 hPa, achieving correlations exceeding 0.57. At 30 hPa and 10 hPa, correlations still reach above 0.37. These results indicate that the EM4 indeed has strong predictive capability for variations in the intensity of the Antarctic polar vortex. The underlying mechanisms are worthy of further analysis.

However, a significant false alarm occurred in 2019, which can be attributed to the rare sudden stratospheric warming (SSW) event in September 2019. This event is marked by a green vertical line in Figure 12.

Due to space limitations, the modes beyond the fourth are not further analyzed in this paper.

## 4. Conclusions

In this study, we applied the EMA to reanalysis data of zonal wind at 70–10 hPa to uncover large-scale climate patterns in the middle stratosphere. Our analysis focused on the first four leading eigen microstates (EM1–EM4), which collectively account for approximately 40–50% of the variance.

The first eigen microstate (EM1) corresponds to the quasi-biennial oscillation (QBO) mode. Its spatial patterns reveal strong coherence in tropical stratospheric zonal winds. Its temporal evolution exhibits a periodic reversal approximately every 2.4 years. EM1’s spatial patterns accurately capture the spatial extent and signal strength of the QBO, while its temporal evolutions exhibit a strong correlation with the QBO indices.

The second eigen microstate (EM2) describes the variations in the Arctic polar vortex. Its spatial patterns are primarily concentrated in the mid-to-high latitudes of the Northern Hemisphere, with strong westerlies around 60° N–90° N and easterlies between 30° N–50° N. Its temporal evolution aligns closely with the deseasonalized Arctic polar vortex index, containing both high-frequency signals and low-frequency trends, including cycles of approximately 1.5 years, 2.4–2.6 years, and around 5–6 years. The 1.5-year cycle is linked to the influence of ENSO, while the 2.4–2.6-year cycle corresponds to the impact of QBO. The 5–6-year cycle in the Arctic polar vortex, which is rarely mentioned in existing research, potentially links to solar and volcanic activities. The EMA can automatically extract modes describing Arctic polar vortex variations from year-round data, making it easier to conduct spectral analysis and significance testing. The 5–6-year cycle of the Arctic polar vortex and its relationship with the North Pacific SST is a topic worthy of further in-depth research.

The third eigen microstate (EM3) characterizes the variations in the Antarctic polar vortex. Its spatial patterns highlight strong westerlies around 60° S. Its temporal evolution closely matches the deseasonalized Antarctic polar vortex index. Unlike EM2 (Arctic polar vortex mode), EM3 lacks a significant 5–6-year cycle, instead displaying an approximately 11-year solar cycle. During winter in each hemisphere, the polar vortex is isolated from solar radiation, with the sun’s influence becoming direct only after spring begins. The Antarctic polar vortex breaks up later than the Arctic one, making it receive more solar radiation during its peak variability in late spring and early summer in the Southern Hemisphere. This leads to a more pronounced 11-year cycle.

The fourth eigen microstate (EM4) has dipole-like spatial patterns concentrated in the mid-to-high latitudes of the Southern Hemisphere. In the lower stratosphere, its temporal evolution is highly correlated with the Niño 3.4 index, while in the middle stratosphere, EM4’s spatial patterns display a wave-1-like pattern. We found that EM4 exerts a leading influence on EM3 by about 2 months, suggesting that EM4 might be one of the factors affecting Antarctic polar vortex variations and could potentially be used to predict its intensity. We designed a linear prediction model that successfully demonstrated its predictive capability.

The EMA is a powerful tool for studying the spatiotemporal evolution of complex systems and has the potential for application to other complex systems as well.

## Figures and Tables

**Figure 1 entropy-27-00327-f001:**
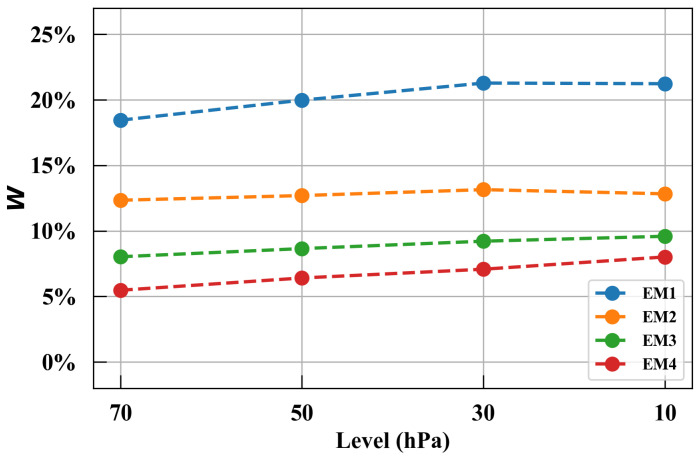
The contributions of the top four eigen microstates (EM1 to EM4) at 70–10 hPa. Blue, orange, green, and red points indicate EM1 to EM4, respectively; derived from ERA5 Reanalysis data (1980–2022; see Section 2).

**Figure 2 entropy-27-00327-f002:**
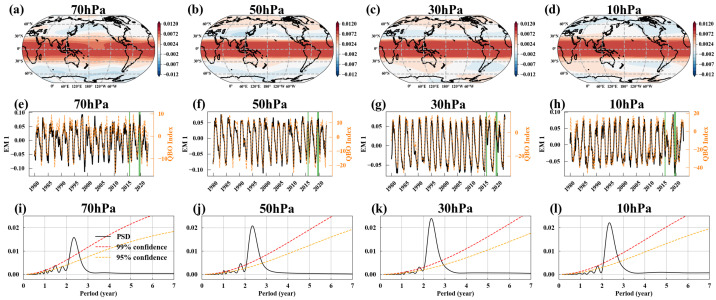
EM1. (**a**–**d**) Spatial patterns of EM1 at 70, 50, 30, and 10 hPa, where positive values (red) indicate westerlies and negative values (blue) indicate easterlies; (**e**–**h**) temporal evolution of EM1 at 70, 50, 30, and 10 hPa (black solid lines) with QBO index (orange dashed lines), showing high correlation (0.88, 0.94, 0.96, 0.95). Green-shaded areas indicate moments when QBO disruptions occurred. (**i**–**l**) Power spectral density of EM1’s temporal evolution at 70, 50, 30, and 10 hPa, with red and orange dashed lines representing 99% and 95% confidence levels.

**Figure 3 entropy-27-00327-f003:**
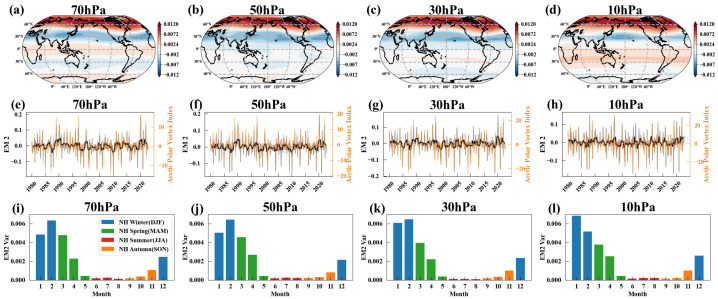
EM2. (**a**–**d**) Spatial patterns of EM2 at 70, 50, 30, and 10 hPa, with positive values (red) indicating westerlies and negative values (blue) indicating easterlies. (**e**–**h**) Temporal evolution of EM2 at 70, 50, 30, and 10 hPa (gray solid lines), with the orange dashed lines representing the deseasonalized Arctic polar vortex index (NPVI) (see Section 2), showing a high degree of alignment, with correlation coefficients of 0.93, 0.92, 0.93, and 0.96, respectively. The black solid line represents the 12-month smoothed temporal evolution of EM2. Panels (**i**–**l**) seasonal variance of EM2 at 70, 50, 30, and 10 hPa, with green, red, orange, and blue representing spring, summer, autumn, and winter in the Northern Hemisphere, respectively.

**Figure 4 entropy-27-00327-f004:**
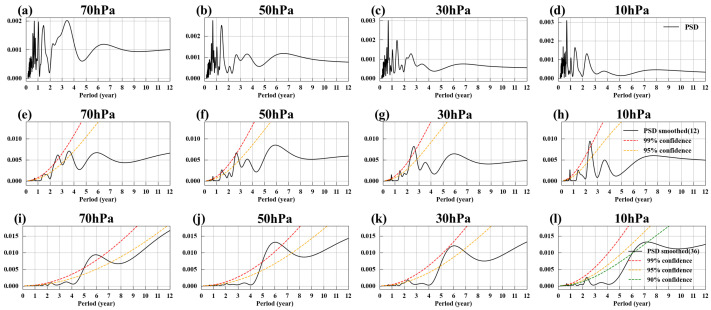
Power spectral density of EM2. (**a**–**d**) Power spectral density of EM2 at 70, 50, 30, and 10 hPa. (**e**–**h**) Power spectral density of EM2 with a 12-month smoothing at 70, 50, 30, and 10 hPa. (**i**–**l**) Same as (**e**–**h**) but with a 36-month smoothing. Red and orange dashed lines represent 99% and 95% confidence levels, respectively; green-dashed line represents 90% confidence level.

**Figure 5 entropy-27-00327-f005:**
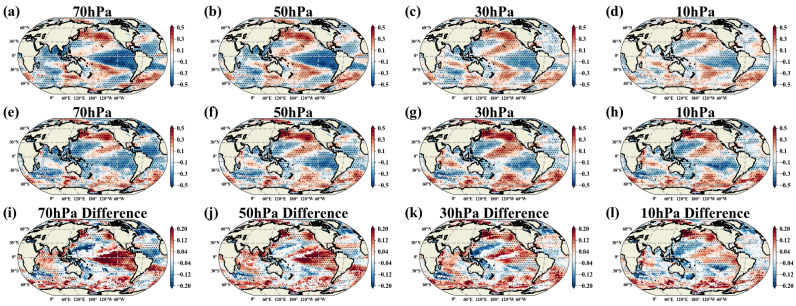
Correlation between EM2 and SST. (**a**–**d**) Correlation coefficients between EM2 temporal evolutions at 70, 50, 30, and 10 hPa and SST (both time series are smoothed over 12 months before the correlation analysis). (**e**–**h**) Same as (**a**–**d**), but with 36-month smoothing. The dashed area indicates regions where the correlation coefficients exceed the 90th percentile threshold. (**i**–**l**) Differences between the 36-month-smoothed correlation and the 12-month-smoothed correlation. Positive values (red) in the figure indicate that the 36-month-smoothed correlation is higher than the 12-month-smoothed correlation. Only regions where the correlations from both smoothing methods share the same sign are shown in the figure; regions with opposite signs are set to 0. The dotted areas indicate regions that are statistically significant in both methods.

**Figure 6 entropy-27-00327-f006:**
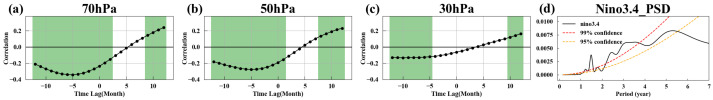
Lead–lag correlation between EM2’s temporal evolutions and the Niño 3.4 index and power spectral density of the Niño 3.4 Index. (**a**–**c**) Lead–lag correlation between EM2 at 70, 50, and 30 hPa and the Niño 3.4 index (both times series are smoothed over 12 months before the correlation analysis); green-shaded areas indicate *p* < 0.01. Lag < 0 indicates the Niño 3.4 index leads EM2. (**d**) Power spectral density of the Niño 3.4 index, with red and orange dashed lines representing 99% and 95% confidence levels, respectively.

**Figure 7 entropy-27-00327-f007:**
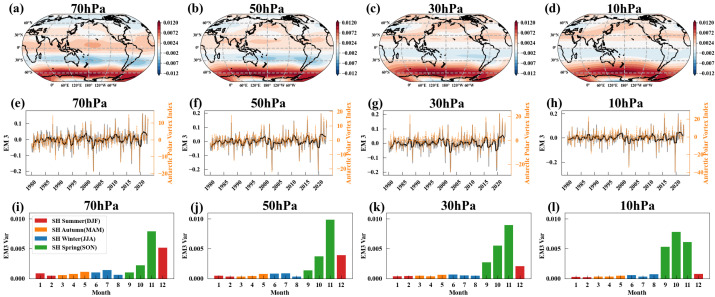
EM3. (**a**–**d**) Spatial patterns of EM3 at 70, 50, 30, and 10 hPa, with positive values (red) indicating westerlies and negative values (blue) indicating easterlies. (**e**–**h**) Temporal evolutions of EM3 at 70, 50, 30, and 10 hPa (gray solid lines), with the orange dashed lines representing the deseasonalized Antarctic polar vortex index (SPVI) (see Section 2), showing a high degree of alignment, with correlation coefficients of 0.88, 0.93, 0.91, and 0.89, respectively. The black solid line represents the 12-month smoothed temporal evolutions. (**i**–**l**) Seasonal Variance of EM3 at 70, 50, 30, and 10 hPa, with green, red, orange, and blue representing spring, summer, autumn, and winter in the Southern Hemisphere, respectively.

**Figure 8 entropy-27-00327-f008:**
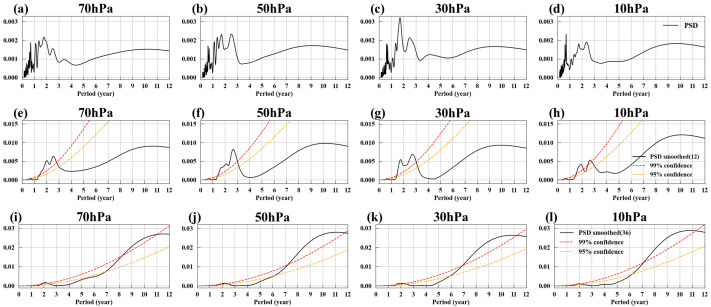
Power spectral density of EM3. (**a**–**d**) Power spectral density of EM3 at 70, 50, 30, and 10 hPa. (**e**–**h**) Power spectral density of EM3 with a 12-month smoothing at 70, 50, 30, and 10 hPa. (**i**–**l**) Same as (**e**–**h**) but with a 36-month smoothing. Red and orange dashed lines represent 99% and 95% confidence levels, respectively.

**Figure 9 entropy-27-00327-f009:**
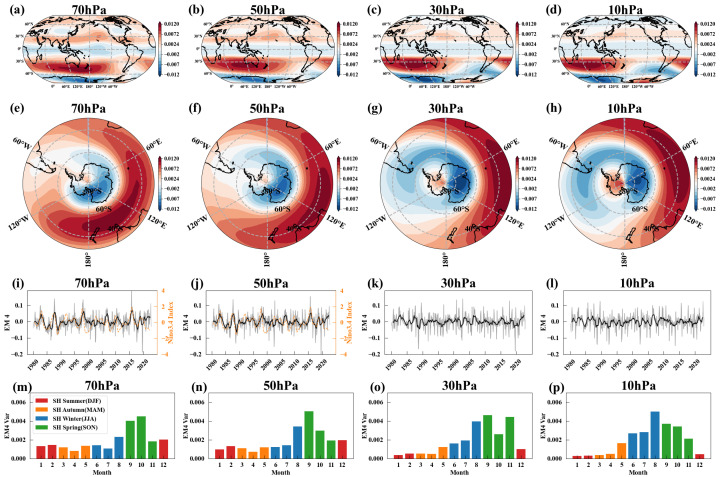
EM4: Southern Hemisphere dipole mode. (**a**–**d**) Spatial patterns of EM4 at 70, 50, 30, and 10 hPa, with positive values (red) indicating westerlies and negative values (blue) indicating easterlies. (**e**–**h**) the same as (**a**–**d**) but in the Antarctic polar projection. (**i**–**l**) Temporal evolutions of EM4 at 70, 50, 30, and 10 hPa (gray solid lines), with the black solid line representing the 12-month smoothed temporal evolutions. (**m**–**p**) Seasonal Variance of EM4 at 70, 50, 30, and 10 hPa, with green, red, orange, and blue representing spring, summer, autumn, and winter in the Southern Hemisphere, respectively.

**Figure 10 entropy-27-00327-f010:**
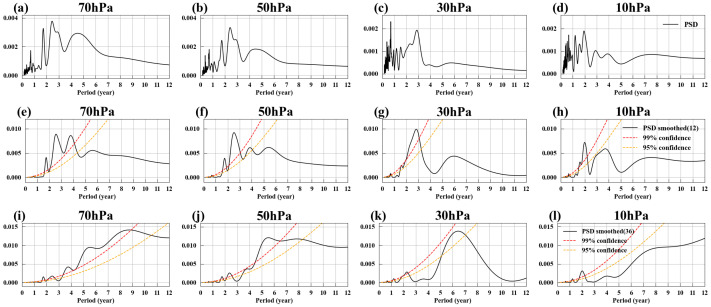
Power spectral density of EM4. (**a**–**d**) Power spectral density of EM4 at 70, 50, 30, and 10 hPa. (**e**–**h**) Power spectral density of EM4 with a 12-month smoothing at 70, 50, 30, and 10 hPa. (**i**–**l**) Same as (**e**–**h**) but with a 36-month smoothing. Red and orange dashed lines represent 99% and 95% confidence levels, respectively.

**Figure 11 entropy-27-00327-f011:**
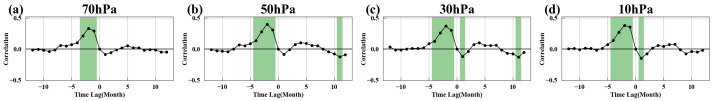
Lead–lag correlation between EM4 and EM3. (**a**–**d**) Results at 70, 50, 30, and 10 hPa, respectively. Time Lag < 0 indicates that EM4 leads EM3. The green-shaded areas indicate *p*< 0.01.

**Figure 12 entropy-27-00327-f012:**
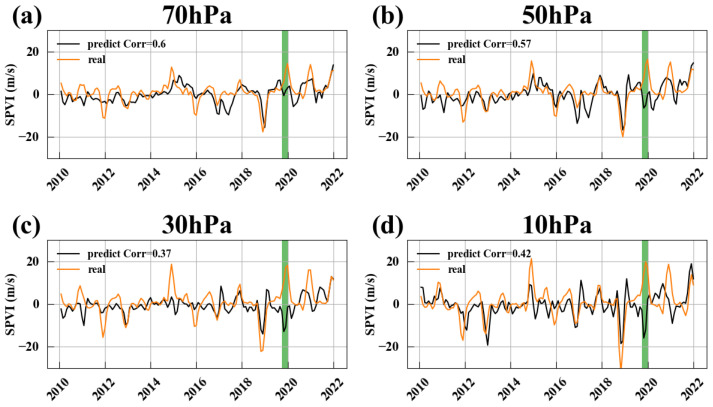
The predictive capability of EM4 for the Antarctic polar vortex: Panels (**a**–**d**) represent the 70 hPa, 50 hPa, 30 hPa, and 10 hPa levels, respectively, with predictive correlations of 0.60, 0.57, 0.37, and 0.42. The green vertical line indicates the time of occurrence of a sudden stratospheric warming event in September 2019.

## Data Availability

The data that support the findings of this study are freely available. The ERA5 reanalysis dataset is available at https://cds.climate.copernicus.eu/datasets/reanalysis-era5-pressure-levels-monthly-means?tab=overview (accessed on 18 March 2025). The Niño 3.4 index provided by NOAA (National Oceanic and Atmospheric Administration) is available at https://psl.noaa.gov/data/correlation/nina34.data (accessed on 18 March 2025).

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
