# Peer review of "A Revisit of Large-Scale Patterns in Middle Stratospheric Circulation Variations"

_entropy, 2025, doi:10.3390/e27040327_

Round 1
Reviewer 1 Report
Comments and Suggestions for Authors
This manuscript ,by my opinion, strongly needs to be published. The reasons are as follows:
- A new advanced method for data processing, based in significant level on already known basic statistical procedures (time series analysis, variance and correlation coefficients calculations etc.)- the so called “Eigen Microstate Approach”(EMA). However it is not enough clear for me why we should consider the EMA as a better technology as for example the standard linear/ non-linear multiple correlation-regression models, where the interaction between the investigated parameters is also taken into account? Is there the principal difference only in the fact of more detailed spatial and/or time resolution, which relates to larger volumes of data and calculations in EMA?
RECOMMENDATION
I could recommended of authors to give a more detailed description (short numerical example) of EMA in an additional “Appendix” section after bottom of basic manuscript content. It could seriously increases the interest to EMA as well as to better citation of the paper.
2.As a very interesting and significant could labeled the establishing of 11yr (obviously solar modulated) cycle in EM3/resp. Antarctic Polar Vortex Variations and ~6yr cycle in EM2/Arctic Polar Vortex (see [1] and the cites therein) the cycles by duration of 5-6 years in natural terrestrial phenomena are by solar activity origin and it correspond to opposite phases of 11yr solar cycle (a significant phase shifting of the relationships is possible). It is interesting to note that during the last few years a number of papers regarding the solar and volcanic activities relationship has been published [2-5]. The general conclusion in these studies is that the volcanic activity is modulated by Sun and corresponding space weather processes and the maximal efficiency of this relationship occurs in two 11yr solar cycle phases – near to sunspot minima and maxima. The relationship is stronger for the strongest volcanic eruptions . Due to this circumstance a 5-6 year cycle in different natural terrwestrial processes is possible, including also in stratosphere and troposphere and on this base – in Earth climate. The possible physical mechanism is the forcing of electrical current systems between ionosphere and upper lithosphere by solar X and UV radiation during the solar maxima variations (extreme events are the solar flares) as well as the galactic cosmic rays (GCR) during the solar minima (see [3-5] and cites therein for details). The total solar irradiance (TSI index) variations influence in this case are negligible.
The both main centers of low atmospheric pressure and cyclonic activity near Iceland and Bering Sea (Kamchatka, Alaska, Aleut Islands ) in North Hemisphere corresponds also to centers of volcanic activity. Potentially it could be related to 5-6 yr cycle activity as well as to Arctic Polar Vortex variations.
RECOMMENDATION
It could/ (should) add a short Section “Discussion” in relation to mentioned above.
- The obtained phase shifted relationship between Southern Hemisphere Dipole Mode and Antarctc Vortex is useful for prediction aims.
TECHNICAL REMARK
The text in and below figures is “too microscopic” and it is difficult to read it. Please, correct the problem.
SUMMARY
The paper is very interesting. It could be seriously improved if the above mentioned recommendations will be done, i.e. after a “moderate” re-edition.
REFERENCES
1. Vitinskii, Y.; Ohl, A.; Sazonov, B. Solnce i Atmosfera Zemli/The Sun and Earth Atmosphere; Gidrometeoizda: Leningrad, USSR, 1976. (In Russian)/ OR SEE ON ENGLISH: Herman J.R. and Goldberg R., 1978, Sun, Weather and Climate, NASA Sci an Technology Inf. Branch
2. Qu, W-Z., Huang, F., Du, L., et al., The periodicity of volcano activity and its reflection in some climate factors, Chin. J. Geophys., 2011, vol. 54, no. 2, pp. 135–149.
3. Komitov, B.P.; Kaftan, V.I. “Danjon Effect”, Solar Activity, and Volcanism. Geomagn. Aeron. 2022, 62, 1117–1122
4. Komitov, B.P.; Kaftan, V.I. The Lower Ionosphere and Tectonic Processes on Earth. Geomagn. Aeron. 2023, 63, 176–184.
5. Komitov, B. About the Possible Solar Nature of the ~200 yr (de Vries/Suess) and ~2000–2500 yr (Hallstadt) Cycles and Their Influences on the Earth’s Climate: The Role of Solar-Triggered Tectonic Processes in General “Sun–Climate” Relationship. Atmosphere 2024, 15, 612.
https://doi.org/10.3390 atmos15050612
Reviewer 2 Report
Comments and Suggestions for Authors
Review of “A Revisit of Large-Scale Patterns in Stratospheric Circulation Variations” by Ningning Tao et al.
Observational and model studies of the stratospheric circulation and its coupling with the troposphere have been performed since the end of the 20th century. The use and development of big data analysis by climate physics and mathematicians contributed to the intensive research and the improvement in scientific understanding of stratospheric circulation and its variability – both natural and anthropogenically forced variability and change.
This study, submitted to the entropy journal, intends to revisit the variations of large scale stratospheric circulation, that is, the authors aim at discussing the four main temporal variability cycles identified by means of the Eigen Microstate Approach (EMA) applied to monthly “zonal wind at 70-10 hPa from 1980 to 2022, based on ERA5 reanalysis data.” While this study might convey corroboration of well know features previously identified with other statistical tools and/or other datasets, in my opinion in its present form it cannot be published before a major revision as it includes serious flaws. My main concerns with this manuscript are below.
11. Atmospheric sciences terminology and technical language – through the text the authors evidence strong deficiencies in the use of scientific and technical terms. For example, in physics of the climate the words “variations” and “change” represent different concepts and scales: the text should be revised, and the authors should clarify if they are referring to time variability or to a definite change. From my understanding this study is not addressing the topic of climate change and thus there should not be this misunderstanding. Another example is that the expressions “Antarctic Polar Vortex, Arctic Polar Vortex, and Quasi-Biennial Oscillation” are not considered “phenomena”. The authors state that “Our analysis focused on the three leading modes, which correspond to the Quasi-Biennial Oscillation (QBO), the Arctic and Antarctic Polar Vortices respectively.” – leading modes of variability have been clearly defined since the end of the 20th century, and they do not correspond to the Polar Vortices. These are few examples that suggest the need to revise the climate concepts and terminology.
22. I believe the readers of Entropy journal are well familiar with the fact that “The atmosphere is composed of several layers: the troposphere, stratosphere, mesosphere, thermosphere, and ionosphere, from the bottom up.” and they will easily revise basic characteristics of the atmosphere. This first sentence from the introduction illustrates the superficiality used in writing the introduction which continued through the presentation and discussion of the results. The authors address lightly many different topics, with a selective choice of literature, without in-depth review of the specific topics that will be discussed in the following sections. There is not a reference to recent reviews on the topic, there is not a comprehensive, substantiated justification of the literature and scientific gap to be addressed in this study, there is not a highlight of the research questions to be addressed in this manuscript.
33. Sentences such as: “Given the limited data, many aspects of the stratosphere could only be studied qualitatively”, “However, previous studies still tends to focus on specific climate phenomena within the stratosphere.”, “The climate system is a highly complex system, and traditional reductionist approaches are insufficient to fully understand it.” are incorrect, and/or trivial, generic and/or outdated. The climate system, as all complex systems, is not fully understood and this is how science advances. It is expected that a scientific article uses objective sentences, with scientific rigour and grounded in literature.
44. It is not true that there is not “a comprehensive view of large-scale stratospheric circulation patterns.” Large scale stratospheric circulation has been studied for decades – namely within the World Climate Research Programme (WCRP) international initiatives such as the “Stratosphere-troposphere Processes And their Role in Climate” to address Atmospheric Dynamics and Predictability, including the solar influence, dynamic variability and changes through the use of long-term climate records.
55. In the end of the introduction the authors state “Given that stratospheric atmospheric motion is primarily dominated by zonal winds, in this paper we will use the EMA to study the large-scale patterns of global stratospheric zonal winds.” This motivation should be addressed in the introduction: what the state of knowledge on the topic is; what the gaps in the literature are; which “traditional reductionist approaches” have been used and why EMA should be used instead – what is the added value of EMA, what does EMA bring to this research topic that was not possible with previous technics. In sum, what the novelty of this study is.
66. Data and methods: the use of monthly data should be justified. The choice of the pressure levels should also be justified. The authors state “ERA5 provides wind speed data (…) to the top of the stratosphere (1 hPa).” It would be expected that all levels above 70 hPa would be used to study “the lower to upper stratosphere”. Either this study is not studying the all stratospheric circulation, or the choice of pressure levels should be better justified. This is justified on section 3.Results, without any references (ln 149-152). This text is another example of the lack of grounding of the manuscript: “it has been indicated that the layers most closely associated with tropospheric weather and climate lie between 70 hPa and 10 hPa. Therefore, our study focuses on the four representative layers: 70, 50, 30, and 10 hPa”. “It has been indicated” – who (references are needed), and for what purposes – the link between the study of “Large-Scale Patterns in Stratospheric Circulation Variations” and the Stratospheric-Tropospheric coupling has not been stated in this manuscript and it is a distinct research topic.
77. Ln 157-158 – “As the pressure level decreases, there is a slight upward trend in the contribution of each mode. This is likely because, with increasing altitude, atmospheric motion patterns simplify, thereby increasing the contribute of dominant patterns.” Please explain what the sentence “there is a slight upward trend in the contribution of each mode” means. Please refer to what the method used to calculate the trend is. Please refer to its statistical significance. Please discuss how these percentages compare to previously published results obtained with other methods. Please avoid using expression such as “This is likely because…” and use published literature to confirm or not these statements. Again, as stated above, the discussion of the results must be based on literature review and the authors must justify why they may refer that “temporal evolutions, obtained from global zonal wind using the EMA, are possibly more suitable as new QBO indices.” In science, either the authors make evidence of which other indices are not more suitable, or these statements cannot be presented.
88. As previously stated, these are topics of intensive research for several decades. The dynamics of the QBO, of the Artic and Antarctic Polar Votex, the influence of the solar radiative forcings on the variability, all these topics have been quantitatively studied and there is much recent literature which should be cited and discussed in this manuscript. The results shown should be described in more detail and should be quantitatively compared with available data published in the literature, so that advance in science may result.
99. To the best of my knowledge, one of the advantages of the EMA is the study of the phase transition of complex systems. This is one of the aspects that I was expecting to see in the discussion of these results.
110. Even though I may agree that “The EMA is a powerful tool for studying the spatiotemporal evolution of complex systems. It not only reveals known patterns, providing new perspectives for describing climate phenomena more accurately, but also uncovers unknown climate phenomena, offering new avenues for identifying hidden patterns within the climate system.”, this sentence is not the conclusion of the present study. Nor this study can confirm this sentence.
In its present form this study is the application of a different tool to well know ERA5 reanalysis data to confirm well know stratospheric circulation features, which cannot make evidence on how it may be a relevant contribution to Entropy journal.
Round 2
Reviewer 1 Report
Comments and Suggestions for Authors
I found too much work , which has been performed by author for a serious improvement of the manuscript. The main recommendations from my side were done in the new version of the text. By my opinion it is now ready for publication.
Reviewer 2 Report
Comments and Suggestions for Authors
The authors have addressed most of the comments, they have reformulated the manuscript and they made comprehensive improvements.
The introduction is now focusing on specific research topics on the stratosphere and on the relevance, novelty and advantage of using the EMA methodology. The refernce list is significantly improved and discussion of results is now made with reference to the literature. I believe the authors have now shown the originality and significance of their study. However, there are still some aspects that need to be improved before the manuscript is accepted for publication.
Data and methods section must be improved.
On section 3.1 the authors write (ln 139-140) “we used the Niño 3.4 index provided by NOAA (National Oceanic and Atmospheric Administration) [74]”. Please describe, similarly, including a reference, all other indices used in this study. For example, on ln 299 it is stated: “represent the deseasonalized Arctic Polar Vortex index”, without a reference to which index and at which level was used – these data and methodology must be described in the “Data and methods” sections, for replicability. A sentence such as “the Arctic Polar Vortex index is based on the average zonal wind at 60°N” in a figure caption is not adequate. Similar comment to (ln 443) the definition of the Antarctic Polar Vortex Index (SPVI).
In addition, all methods used should be included in the Data and Methods section. This applies as well to the correlation methods (it is not adequate to describe only a method in a figure caption, as for example in figure 5) and also to the methodology to develop a model. It is useful to have the criteria used in the Monte Carlo technic near the figure, but a mention to the methodology and the adequate references should be added.
Figure 2 description and analysis – the reference to “Green shaded areas indicate moments when QBO disruptions occur” is only made in the Figure 2 caption. A paragraph on this should be included in section 3.1. The authors have mentioned, on their response to reviewers, that this will be a topic of a future work. Nevertheless, some details should be given to how the “moments when QBO disruptions occur” are computed and identified; what they mean and specifically how they correspond to reality in figure 2; why there are no other “QBO disruptions” identified through the time series – through eye analysis many other occurences could be identified previous to 2015. It should be discussed comprehensively in Figure 2 analysis, even if as future research.
Please discuss whether there are no disruptions when analyzing the signals for Artic and Antarctic Polar Vortex Circulations. As previously mentioned, this might be a strong advantage of this research and it should be highlighted.
Please replace the word “chapter” by “section”
Ln 139 please correct “covering the lower and middle stratosphere”
Ln 343 – please remove the personal pronoun “he” – usually one should use the expressions “the author” or “the authors”. Please be aware if Dr. Jevrejeva is a male or female researcher.
Figure 5 – to better highlight the differences, it would be useful to include a 3rd row in this figure with the difference between both panels at the same level.
Ln 366 – please remove the expression “During the review process, the reviewers proposed”. In my opinion, this is not adequate in a research paper. The responsibility of the work and the text is of the authors and, in my opinion, the reviewers – specially when they don’t sign their comments – should not be attributed to the sentences and results analysis. Additionally, the authors may include in the acknowledgments section a sentence such as “the authors thank the comments made by the reviewer that contributed t improve the manuscript.” The authors should rephrase this and other similar sentences and discussion using adequate references and avoiding sentences that may be speculative. As the authors well know, a statistically significant correlation is not synonyms to a causal relation without a suggested mechanism is advanced.
The results identified in this study (ln 361-364 and ln 411-413), as the authors mention, are not conclusive and need further research that may confirm their justification or not, namely through dedicated numerical modeling research. The dynamics of the Artic and the Antarctic Polar Vortices are intrinsically very different, and the authors have not shown that there is a link between “the Antarctic Polar Vortex is more susceptible to solar radiation” and a 11-year cycle. In this present study the results do not present the mechanisms that may justify that the 6-yr signal corresponds to a real physical mechanism in the climate system. These aspects may be suggested, but the phrasing must be adjusted.
Ln 418 – please replace “inside and outside this boundary” for “north and south of this parallel”
Figure 11 – please avoid repetitions in the caption.
Ln 442 to 453 – this text should be included in the Data and Methods section, and here the authors should only present and discuss the model. The authors should discuss how relevant and useful is a linear model to predict a complex behavior of the Antarctic Polar Vortex. Figure B1 should be presented and discussed as Figure 12.
Ln 507-510 – please rephrase the sentence as the results shown do not establish this causal relationship.
Ln 513 Please revise the text, throughout the manuscript, avoiding contradictory sentenses on 30hPa and 10hPa to correspond to the upper stratosphere
Round 3
Reviewer 2 Report
Comments and Suggestions for Authors
I can recommend this revised manuscript for publication. I thank the authors for their constructive approach throughout the review process.
The Physics of Climate is generally studied by scientists who have a strong background in Physics and apply it to studying and advancing knowledge on the complex Climate System. I never sign my reviews, as I consider there should not be any misunderstanding on the voluntary and altruism of the review process for the advance of Science. However, in this case - even though I want my review to remain anonymous - I would like to add some additional thoughts.
The authors decided to submit their study to a journal named "Entropy". Thus, I would like to call your attention to the paper by Professor José Pinto Peixoto, one of the pioneers on the study of Information Theory and Entropy.
Peixoto, J. P., Oort, A. H., De Almeida, M., Tomé A.: Entropy budget of the atmosphere, Journal of Geophysical Research: Atmospheres, 96, 10981-10988, https://doi.org/10.1029/91JD00721, 1991.
Peixoto was a Professor at the University of Lisbon and MIT and he is coauthor of the classic book on Physics of Climate (1992; https://www.osti.gov/biblio/7287064). He was also my Professor and he made school in the energetics of the atmosphere in the University of Lisbon.
In case you wish to porsue your study and advance now on the dynamical mechanisms of the variability of the atmosphere and the climate system, I would like to also suggest the following articles on the variability of the stratospheric polar vortex:
Liberato, M. L. R., Castanheira, J. M., de la Torre, L., DaCamara,C. C., Gimeno, L.: Wave energy associated with the variability of the stratospheric polar vortex, Journal of the Atmospheric Sciences, 64 (7), 2683-2694, https://doi.org/10.1175/JAS3978.1, 2007.
Castanheira, J. M., Liberato, M. L. R., de la Torre, L., Graf, H-F., and DaCamara, C. C.: Baroclinic Rossby Wave Forcing and Barotropic Rossby Wave Response to Stratospheric Vortex Variability, Journal of the Atmospheric Sciences, 66 (4), 902–914, https://doi.org/10.1175/2008JAS2862.1, 2009.
Castanheira, J. M., Añel, J. A., Marques, C. A. F., Antuña, J. C., Liberato, M. L. R., de la Torre, L., and Gimeno, L.: Increase of upper troposphere/lower stratosphere wave baroclinicity during the second half of the 20th century, Atmos. Chem. Phys., 9, 9143–9153, https://doi.org/10.5194/acp-9-9143-2009, 2009.
